# The geometry of hidden representations of large transformer models

**Lucrezia Valeriani**[1,2*]     **Diego Doimo**[1,3*]     **Francesca Cuturello**[1]

**Alessandro Laio**[3,4]     **Alessio Ansuini**[1†]     **Alberto Cazzaniga**[1†]

[1] AREA Science Park, Trieste, Italy
[2] University of Trieste, Trieste, Italy
[3] SISSA, Trieste, Italy
[4] ICTP, Trieste, Italy

## Abstract

Large transformers are powerful architectures used for self-supervised data analysis across various data types, including protein sequences, images, and text. In these models, the semantic structure of the dataset emerges from a sequence of transformations between one representation and the next. We characterize the geometric and statistical properties of these representations and how they change as we move through the layers. By analyzing the intrinsic dimension (ID) and neighbor composition, we find that the representations evolve similarly in transformers trained on protein language tasks and image reconstruction tasks. In the first layers, the data manifold expands, becoming high-dimensional, and then contracts significantly in the intermediate layers. In the last part of the model, the ID remains approximately constant or forms a second shallow peak. We show that the semantic information of the dataset is better expressed at the end of the first peak, and this phenomenon can be observed across many models trained on diverse datasets. Based on our findings, we point out an explicit strategy to identify, without supervision, the layers that maximize semantic content: representations at intermediate layers corresponding to a relative minimum of the ID profile are more suitable for downstream learning tasks.

## 1 Introduction

Deep learning has significantly changed the landscape of scientific research and technology advancements across multiple disciplines in recent years. A particular class of deep learning models, transformers trained with self-supervision, combine high predictive performance and architectural simplicity. These transformer models consist of a stack of identical self-attention blocks trained in a self-supervised manner using masked language modeling (MLM) or auto-regressive objectives [1, 2]. It has been shown that the features learned by these models can be used to solve a wide range of downstream tasks in natural language processing [3–5], biology [6–9], and computer vision [10–12].

Previous studies analyzing convolutional architectures have shown that data representations in deep learning models undergo profound changes across the layers [13, 14]. In transformers, each module maps the data into a representation, and it has already been observed that the organization of representations in the last hidden layer can reflect abstract, domain-specific properties [10]. However,

---

*These authors contributed equally to this work.

†Correspondence: alessio.ansuini@areasciencepark.it, alberto.cazzaniga@areasciencepark.it

37th Conference on Neural Information Processing Systems (NeurIPS 2023).

in models trained by self-supervision, the last representation generally has the role of allowing reconstruction of the input representation. Therefore, the most semantically rich representation is likely to arise within the intermediate hidden layers of the network.

In this work, we study the intrinsic dimension and neighbor composition of the data representation in ESM-2 [15] and Image GPT (iGPT) [10], two families of transformer architectures trained with self-supervision on protein datasets and images (see Sec. 2.3). We find consistent within-domain behaviors and highlight similarities and differences across the two domains. Additionally, we develop an unsupervised strategy to single out layers carrying the most semantically meaningful representations.

Our main results are:

- representations in large transformers evolve across the layers in distinct phases, revealed by simultaneous changes in global ID and local neighborhood composition (Sec. 3.1);

- a common trait of the ID profiles is a prominent peak of the ID, in which the neighbor structure is profoundly rearranged, followed by relatively low ID regions in which the representations encode abstract semantic information, such as the class label for images, and remote homology for proteins (Sec. 3.2). This abstract information emerges gradually while the ID decreases (Sec. 3.1);

- in the case of protein language models (pLMs) and iGPT, our experiments show that the ID profile can be used to identify in an unsupervised manner the layers encoding the optimal representation for downstream learning tasks, such as protein homology searches or image classification (Sec. 3.2).

These findings suggest a general computational strategy learned by these models, in which the reconstruction task is performed in three phases: 1) a data expansion in a high intrinsic dimension space. This expansion resembles the strategy followed in kernel methods [16], where one implicitly expands the feature space by introducing non-linear functions of the input features; 2) a compression phase that projects the data in low-dimensional, semantically meaningful representations. Strikingly, this representation emerges spontaneously in all the self-supervision tasks we considered; 3) a decoding phase where the model addresses the minute decision-making needed to reconstruct the data from a compressed representation. The layers performing this task behave similarly to the decoder layers in a "vanilla" autoencoder. In particular, understanding this behavior can be exploited as an unsupervised method to identify the most semantically rich representations of transformers trained on large datasets where the annotation is scarce.

## 2 Methods

### 2.1 Intrinsic dimension.

Learning complex functions in high-dimensional spaces is challenging because as the dimension of the space increases, the required number of samples must grow exponentially [17]. Fortunately, real-world datasets, such as images and protein sequences exhibit regularities that result in strong constraints and correlations among their features [18, 19]. Consequently, although these datasets may have many features, they lie around low-dimensional, often highly non-linear manifolds. A dataset's intrinsic dimension (ID) is the dimensionality of the manifold *approximating* the data and can be described as the minimum number of coordinates that allow specifying a data point approximately without information loss [20, 21]. We measure the ID with the TwoNN estimator [21], which requires only the knowledge of the distances $r_{i1}$, $r_{i2}$ between each point $x_i$ and its first two nearest neighbors. It can be shown that under the assumption of locally constant density, the ratio $\mu_i = r_{i2}/r_{i1}$ follows a Pareto distribution with a shape parameter equal to the ID. The intrinsic dimension can be inferred by maximum likelihood or linear regression from the cumulative distribution function (see Appendix Sec. B). The algorithm is robust to changes in curvature and density of the data and has been used effectively to analyze representations in deep neural networks in [13, 14] and more recently in [22–24]. We adopt the TwoNN implementation of DADApy [25] and discuss the technical details of our analysis together with the validation of the constant density assumption in Appendix B.

## 2.2 Neighborhood overlap.

Changes in the data representation between successive layers of a model can be traced in the rearrangement of the neighbor structure of each data point. The neighborhood overlap $\chi_k^{l,m}$ introduced in [14] measures the similarity between two representations $l, m$ with the average fraction of common $k$-nearest neighbors of a data point. Consider the activation vector $x_i^l$ of an example at layer $l$. Let $A^l$ be the adjacency matrix with entries equal $A_{ij}^l = 1$ if $x_j^l$ belongs to the first $k$-nearest neighbors of $x_i^l$ in Euclidean distance and 0 otherwise. Then the neighborhood overlap between representations $l$ and $m$ can be written as $\chi_k^{l,m} = \frac{1}{N}\sum_i \frac{1}{k}\sum_j A^l{}_{ij} A_{ij}^m$, where $A^l{}_{ij} A_{ij}^m$ is the number of common $k$-neighbors of point $x_i$ between these representations.

In the presence of a labeled dataset $\{(x_i, y_i)\}$, a generalization of the neighborhood overlap measuring the consistency of a label within the nearest neighbors of each data point can be used as a *semantic probe*. In this case, the "adjacency matrix" with respect to the dataset labels is $A_{ij}^{gt} = 1$ if $y_i = y_j$ and 0 otherwise, and we call $\chi_k^{l,gt}$ overlap with the ground truth classes. Both $\chi_k^{l,m}$ and $\chi_k^{l,gt}$ lie in $[0, 1]$ and depend on the choice of neighborhood size $k$. However, in the Appendix (see Fig. S5), we show that the qualitative trend of the overlap is robust to changes in $k$. We use $k = 10$ when analyzing the overlap with the protein superfamily of the SCOPe dataset, and $k = 30$ in the case of the transformers trained on the ImageNet dataset, consistently with the value used in Doimo et al. [14].

## 2.3 Models and Datasets

**Transformer models for protein and image datasets.** The ESM2 single-sequence protein language models (pLMs) and iGPT networks are characterized by similar architectures. The token vocabulary comprises an alphabet of $\simeq 20$ amino acids for pLMs and 512 colors for iGPTs (see Sec. A and Chen et al. [10] for further details). An embedding layer encodes each sequence token into a vector of size $d$, which is then added to a learned positional embedding common to all sequences. A sequence of $l$ embedded tokens of size $\mathbb{R}^{l \times d}$ is thus processed by a stack of architecturally identical blocks composed of a self-attention map followed by a multi-layer perceptron (MLP) producing a series of data representations of size $\mathbb{R}^{l \times d}$. We extract the hidden representations of the sequences after the first normalization layer of each block and then average pool along the sequence dimension to reduce a sequence into a data point embedded in $\mathbb{R}^d$. This way, we can compute the Euclidean distances needed for the ID and overlap analyses between sequences of different lengths.

We study models pre-trained on large corpora of proteins and images with different self-supervised objectives: ESM2 were trained by Lin et al. [15] on the Uniref50 dataset with a masked language model objective and have been publicly released at github.com/facebookresearch/esm; the iGPT networks were trained by Chen et al. [10] with a next token prediction objective on the ImageNet dataset and are available at github.com/openai/image-gpt.

**Datasets.** We consider two benchmark datasets for our analysis of pLMs: ProteinNet and SCOPe. ProteinNet [26] is a dataset of $25,299$ protein sequences for evaluating the relationships between protein sequences and their structures. We use the ProteinNet training set to analyze the ID curves and compute the neighborhood overlap of consecutive layers. The Astral SCOPe v2.08 (SCOPe) dataset [27] contains subsets of genetic domain sequences hierarchically classified into fold, superfamily, and family. We first preprocess the sequences with the filtering procedure recommended in [28]. Secondly, to define the remote homology task, we select proteins that belong to superfamilies with at least ten members and ensure that each superfamily consists of at least two families. As a result, we obtain a dataset of 10,256 sequences and 288 superfamilies, used to evaluate the $\chi^{l,gt}$ in Fig. 4.

For the analysis of the iGPT representations, we choose $90,000$ images from the ImageNet training set [29]. We randomly select 300 classes and keep 300 images per class.

**Reproducibility.** All experiments are performed on a machine with 2 Intel(R) Xeon(R) Gold 6226 processors, 256GB of RAM, and 2 Nvidia V100 GPUs with 32GB memory. We provide code to reproduce our experiments and our analysis online at github.com/diegodoimo/geometry_representations.

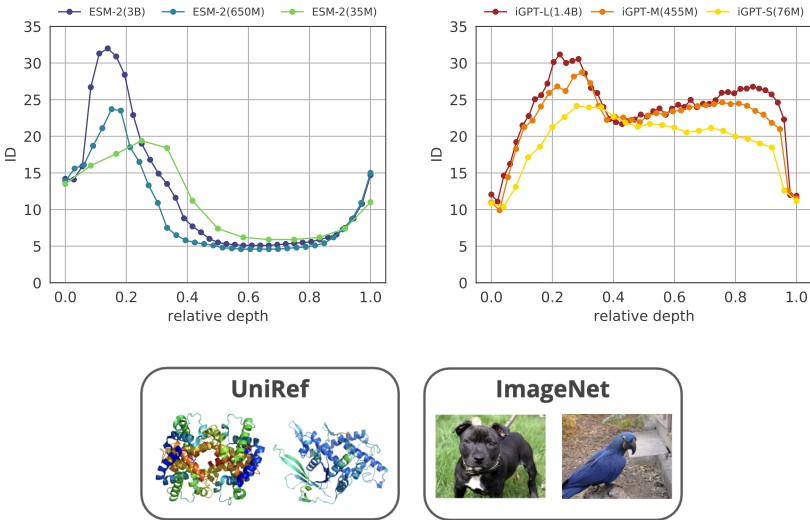

Figure 1: **Intrinsic dimension (ID) of the data representations in hidden layers of large transformers**. (left) The ID profile of ESM-2 protein language models of small (35M, green), medium (650M, blue), and large (3B, purple) sizes show a peak in the first part of the network, a long plateau, and a final ascent. (right) The ID profile in iGPT models of small (76M, yellow), medium (455M, orange), and large (1.4B, red) sizes also present an early peak followed by a less pronounced second peak for the medium and large model. The ID is plotted against the relative depth: the block number divided by the total number of blocks.

## 3 Results

### 3.1 The intrinsic dimension profile in large transformers is similar in different tasks and architectures

In large transformers, the input representation is processed in a sequence of identical blocks, creating a series of representations in vector spaces of the same dimension but radically different in their geometrical properties and semantic content. We start by estimating the ID of each representation for large transformers trained by self-supervision on protein language and image reconstruction tasks. The ID profiles provide a glimpse into the geometry of the manifolds in which the data are embedded. As we will see, the ID changes through the layers in a highly non-trivial manner, with three phases that can be recognized in all the transformers and a fourth phase that we observed only in the iGPT models. In the following sections, we will see how these phases signal the development of structured semantic content in intermediate representations, which are hidden instead in the input and, by construction, in the final embeddings.

**ID curve of protein language models.** We extract representations of the protein sequences in the ProteinNet dataset as explained in Sec. 2.3 and plot their ID against the block number divided by the total number of blocks, referred to as relative depth in Fig. 1. This section describes our findings for pLMs in the ESM-2 model family with 35M, 650M, and 3B parameters. Analogous results hold for other models trained on different datasets, as shown in the Appendix (see Fig. S1).

Figure 1 (left) shows the typical shape of the ID profile in pLMs with three distinct phases: a peak phase, a plateau phase, and a final ascent phase. The peak develops in early layers, spanning roughly the first third of the network. During this phase, the ID expands rapidly, reaching a maximum of approximately 20, 25, and 32 for ESM-2 35M, 650M, and 3B, respectively. The ID then contracts quickly, stabilizing at notably low values, between 5 and 7, that define the plateau phase. While the ID at the peak grows with the model size, during the plateau phase, we observe a quantitative solid consensus in the ID, especially in the layer at the elbow before the final ascent, independent from the embedding dimension that varies from 480 in ESM-2 35B to 2560 in ESM-2 3B. Notably, the ID observed in the plateau phase is also consistent with the values between 6 and 12 measured by

Facco et al. [30]. Their study involved different metrics for computing distances directly applied to pairwise alignments of protein sequences. This remarkable correspondence between different analytical approaches suggests that hidden representations in the plateau region can effectively gauge the underlying degree of variability arising from evolutionary changes in protein sequences. In the final ascent, the ID grows again, returning progressively to values close to the ID computed on the input representation after the positional embedding. This is a consequence of the masked language modeling objective, which focuses on using the context information to recover the missing tokens.

**ID curve of image transformers.** Figure 1 (right) shows the ID profiles of iGPT models of increasing size. Specifically, we examine the small (S), medium (M), and large (L) versions of iGPT trained on ImageNet, which have 76M, 455M, and 1.4B parameters, respectively.

In all cases, the IDs of the output are similar to those of the input, which aligns with the findings from protein language models. In the iGPT-S network, the ID profile has a hunchback shape quite similar to that in convolutional models trained on ImageNet [13] but with a peak value significantly smaller (around 25). This discrepancy could be partly attributed to the different embedding dimensions: equal to 512 in the iGPT-S transformer, whereas in convolutional architectures, it fluctuates across layers and is considerably larger, typically falling within the range of $O(10^4)$ to $O(10^6)$. As we increase the model size, similar to what is observed (pLMs), the maximum intrinsic dimension (ID) grows, reaching 28 for iGPT-M and 32 for iGPT-L. However, after an initial decrease between 0.3 and 0.4 of the relative depth, the ID shows a local minimum of 22 common between the architectures. In the last part of the network, in contrast with what is observed in pLMs, the ID experiences another, more gradual increase, forming a second, shallower peak near the end of the network. Interestingly, the ID of the most compressed representations at around 0.4 of relative depth is compatible with the values observed by Ansuini et al. [13] at the output of a broad range of convolutional classifiers trained on the ImageNet dataset. In their study, the ID varied from 13 to 24 and was found to be correlated with the classifier's accuracy (refer to Figure 4 in Section 3.3 of [13]).

The results shown in Fig. 1 hint at a scenario that will be validated in the following: the ID is an indicator of semantic complexity, and the representations that resemble more closely those extracted by supervised learning on the same datasets are expected to correspond to the relative minimum of the ID curve, situated after the initial peak. This minimum is well defined in iGPTs, while in pLMs, the ID profile forms a plateau.

**The neighborhood rearrangement across the layers mirrors the ID profile.** We will now examine the evolution of the data distribution across the layers, specifically by analyzing the neighborhood overlap $\chi_k^{l,l+1}$, defined in Sec. 2.2. This parameter quantifies the rate at which the composition of the neighborhood of each data point changes between nearby blocks. In deep convolutional models, this rate is significant only in the very last layers [14], where the representation is forced to comply with the ground-truth classification of the images.

In pLMs, we evaluate the neighborhood overlap on the representations of the ProteinNet dataset. The results of our analysis, presented in Fig. 2 (left), reveal a relationship between changes in neighborhood composition and the ID. This observation points to an interesting connection between ID and neighborhood overlap, which differs from what is typically found in convolutional architectures. In the first 40% of the layers, $\chi^{l,l+1}$ remains approximately at 0.5, meaning that the neighborhood composition undergoes substantial changes in each layer, significantly altering representations in the initial part of the network. This corresponds with the peak phase of the ID. However, during the plateau phase, the rate of representation evolution is much slower, with over 90% of the neighbors shared between consecutive layers in the larger models. In the smallest model, characterized by a higher perplexity, the neighborhood composition in the plateau phase is less consistent, with some rearrangements also occurring in successive layers.

In image transformers, Fig. 2 (right), the profiles are qualitatively similar to those observed in protein language models. However, the transition between the initial stage, coinciding with the first ID peak and relative depth $< 0.4$, with rapid changes in neighborhood composition ($\chi^{l,l+1} \sim 0.7$) and a second stage where the rearrangements are less pronounced ($\chi^{l,l+1} \sim 0.9$) is more gradual than pLMs. A significant neighborhood rearrangement is always observed in the last layers, where the reconstruction task is carried on. Similarly to pLMs, $\chi^{l,l+1}$ is lower in shallower models that need faster rearrangements to achieve consistent results in fewer blocks.

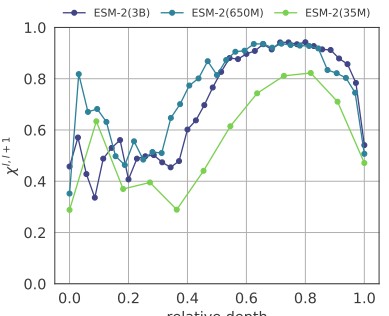 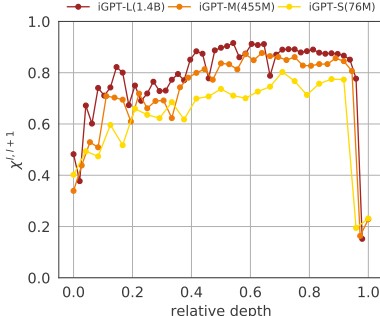

Figure 2: **The neighborhood rearrangement across the layers in large transformers**. (left) Neighborhood overlap of consecutive layers $\chi^{l,l+1}$, computed for ESM-2 (35M, green), ESM-2 (650M, blue), and ESM-2 (3B, purple), shows major local rearrangements in the layers corresponding to the peak and in the final layers and minor changes in the plateau region, mirroring the behavior of the ID. (right) The trend of $\chi^{l,l+1}$ in iGPT-S (yellow), iGPT-M (orange), and iGPT-L (red) is similar to pLMs: the neighborhood composition changes more in early layers where the ID reaches a peak and in the final layers.

**The evolution of the representations during training.** Until now, our focus has been on observing changes in some geometric quantities across layers in fully trained models. We will now focus on understanding how these quantities evolve during training. Specifically, we investigate how representations of the data manifold transform during training for two models: the protein language model ESM-2 (650M)[3] and the image transformer iGPT-L.

For the ESM-2 (650M) model, as shown in Figure 3 (left), we analyze checkpoints corresponding to $[0, 0.1, 0.3, 1, 5] \cdot 10^5$ training iterations, with the last checkpoint representing the fully converged model. In the initial stages of training, we observe a rapid formation of a peak in the early layers, while the ID curve in the remaining part of the network closely resembles that of an untrained model. Between $1 \cdot 10^4$ and $3 \cdot 10^4$ training steps, the ID of the plateau layers substantially decreases. In contrast, the ID measured in the last layers progressively increases towards the input ID. From $3 \cdot 10^4$ training steps to convergence, the ID curve takes its final shape, with a slight increase of the ID measured at the peak and a minor compression at the plateau. The final ID curve is essentially achieved in two stages: first, the initial ID peak emerges, and only in a second phase during training the data representation in the plateau layers is compressed to a lower dimension. This compression is tightly related to the emergence of semantic information that we discuss in Sec. 3.2.

For iGPT-L, we consider the representations at checkpoints corresponding to $[0, 0.13, 0.26, 0.52, 1] \cdot 10^6$ training iterations (see Fig. 3 (right)). The ID dynamics is similar to that of pLMs: before training, the ID curve is flat, and in early training iterations ($< 1.3 \cdot 10^5$), the first ID peak emerges in the initial layers of the network. In the later stages of training, the ID slightly increases at the first peak and decreases at $0.4$ of relative depth, forming the local minimum. Additionally, a second peak emerges in the last third of the model's hidden layers.

## 3.2 The intermediate representations of large transformers are the most semantically rich

A common characteristic among the ID profiles shown in Fig. 1 is the presence of a distinct peak in the first part of the network, observed consistently in all the models we examined. In protein language models, this initial peak is followed by an extended plateau that spans more than half of the layers. In contrast, in iGPT models, a second shallow peak in the ID is observed, with the exception of the iGPT-S model, which does not have one. In section 3.1, we found that the ID at the minimum point between the two peaks is quantitatively similar to the one measured at the output of convolutional networks trained with supervision on the same dataset [13]. This points to a scenario where the representations at this local minimum encode semantic information about the datasets. Chen et al. [10] have already demonstrated with linear probes that features extracted from the intermediate layers

---

[3]The weights of the checkpoints of ESM-2 (650M) analyzed in Fig. 3 were kindly provided by R. Rao.

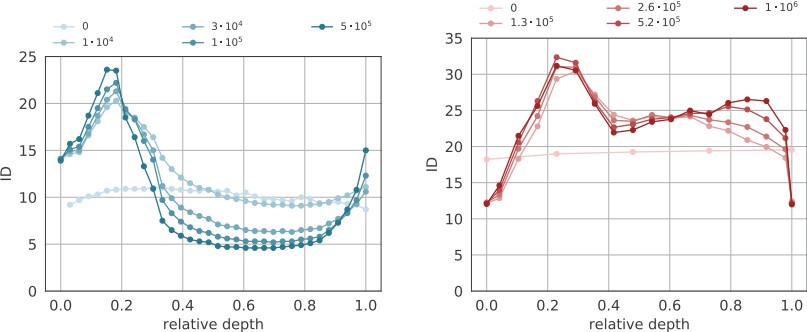

Figure 3: **Evolution of ID curves during training**. (left) ID curves for the pLM ESM-2 (650M) at checkpoints corresponding to $[0, 10^4, 3 \cdot 10^4, 10^5, 5 \cdot 10^5]$ training steps. Firstly, the peak develops; then, the plateau reaches low-ID values, and the ID in the final layers reaches values similar to the input. (right) ID curves for iGPT-L (1.4B) at checkpoints $[0, 0.13, 0.26, 0.52, 1] \cdot 10^6$ training steps. Firstly, the ID peak followed by a plateau with a local low-ID minimum develops; later in training, a second peak emerges in the final layers.

of iGPT models encode this semantic information. In this section, we extend this analysis to protein language models, showing that, in this case, semantic information is maximally present in the whole plateau region of the network. Therefore, we show that the ID profile can be used to predict the specific layers where this information is most pronounced.

**The representations in the plateau region code remote homology relationships.**    Proteins are considered remote homologs when they share a similar structure resulting from common ancestry despite having highly dissimilar amino acid sequences. Rives et al. [6] observed that the Euclidean distances among representations in the last hidden layer of pLMs encode information related to remote homology. We study how this biological feature emerges in pLMs, analyzing the hidden representations of the SCOPe dataset. For every layer $l$, we compute the neighborhood overlap with the *ground truth labels* $\chi_k^{l,gt}$ with $k = 10$. Here, "gt" represents the classification by superfamily, with the exclusion of neighbors in the same family, allowing us to focus on remote homology specifically. Figure 4 (left) shows that structural homology information is absent in the positional embedding layer; it grows smoothly in the peak phase, reaching a stationary maximum $\chi_k^{l,gt} \sim 0.8$ in the plateau phase, and it suddenly decreases in the final ascent phase. This analysis shows that homology information is highest in the plateau layers, and it appears early in the networks right after the peak of the ID. Notably, the predictive power of remote homology by nearest neighbor search is considerably lower in the last hidden layer, scoring at 0.4 instead of 0.8. Therefore, searching for the closest homologs in the plateau layers where protein relationships are better expressed can improve state-of-the-art methods based on representations from the last hidden layer. The Appendix shows that the entity of such improvement is approximately 6% (see Fig. S2).

**Image semantic features emerge in the layer where the ID has a relative minimum.**    It has been shown that predicting pixels in a self-supervised learning context can generate representations that encode semantic concepts [10]. Expanding upon these findings, we establish a connection between this remarkable property and the geometric aspects of representations. Similar to the case of pLMs, we quantify the semantic content of a representation by computing the overlap with the ground truth labels, $\chi^{l,gt}$. In Figure 4 (right), we plot for the iGPT models the overlap with the ImageNet labels for 300 classes of the training set (see Sec 2.3). In all the cases, we consistently observe a peak of $\chi^{l,gt}$ around a relative depth of $\sim 0.4$ where the ID in the larger models iGPT-M and iGPT-L is lower (see Fig. 1). This peak becomes sharper, and the overlap values increase as we scale up the model size. Specifically, the peak value of $\chi^{l,gt}$ is 0.15 in iGPT-S, 0.27 in iGPT-M, and 0.35 in iGPT-L. Similar to our observations in pLMs, the representations where the semantic abstractions are better encoded are also those where the ID is low.

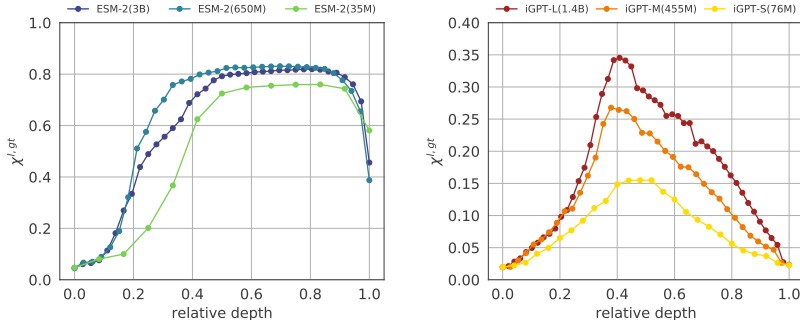

Figure 4: **Local geometry and semantic relations in hidden representations**. (left) Overlap with remote homologs $\chi^{l,gt}$ for ESM-2 (35M, green), ESM-2 (650M, blue), and ESM-2 (3B, purple). Information about remote homologs is highest in the plateau layers, corresponding to low ID values. (right) Overlap with ImageNet labels $\chi^{l,gt}$ for iGPT-S (yellow), iGPT-M (orange), and iGPT-L (red). The information related to ImageNet labels is highest in correspondence with the ID minimum.

# 4   Discussion and conclusions

In this work, we investigate the evolution of fundamental geometric properties of representations, such as the intrinsic dimension and the neighbor structure, across the layers of large transformer models. The goal of this analysis is twofold: 1) understanding the computational strategies developed by these models to solve the self-supervised reconstruction task, and 2) monitoring how high-level, abstract information, such as remote homology in proteins and class labels in images, is related to the geometry of the representations. In particular, we propose an explicit, unsupervised strategy to identify the most semantically rich representations of transformers.

The qualitative picture that emerges from our results is that large transformers behave essentially like sophisticated autoencoders, in which the data are first encoded into a low-dimensional and abstract representation and are successively decoded from it. In the largest iGPT model, this similarity is particularly evident: a second peak in the ID profile is present, approximately mirroring the first, and the overlap with the ground truth labels also varies in an almost symmetric manner in the first and the second half of the network (see Fig. 4 (right)). This makes this model akin to a symmetric autoencoder, in which the decoder performs operations on the representations that are dual to those performed by the encoder.

Crucially, the compression is preceded by an intrinsic dimensionality expansion and a relatively fast-paced rearrangement of the neighbors during the encoding process, which is very similar across all the models we studied. The relation between a low intrinsic dimension at the end of the encoding part and the richness in abstract content of representations is robust and quantitative in all the models we considered. A vast rearrangement of the neighbors in the layers close to the output is also observed for all the architectures we analyze. This is devoted to the process of dealing with the decoding task, and it causes a degradation of the abstract content.

The analysis we performed could be further reinforced, for example, by analyzing the similarity of representations in the ID minima with those generated in models trained in a supervised setting. This can provide a deeper understanding of the role of the second peak in the large iGPT model and the reasons for its appearance in later training phases, as well as offer a clearer interpretation of low ID representations in the plateau phase of pLMs.

**Impact of some implementation choices.**   In this study, we average pool the representations along the sequence dimension as a convenient method to compute the Euclidean distances between proteins of varying lengths. This averaged representation can serve multiple purposes, including measurement of representation quality using linear probes [10], and can provide sufficient biological information to address homology, structural, and evolutionary tasks directly or after fine-tuning. It's worth noting, however, that other approaches for comparing sequences of variable lengths have been explored, such as those in [31] and [22]. When the inputs have all the same sequence length as in

the iGPT case, reducing the sequence representation to a $d$-dimensional vector is also convenient from a computational perspective. Indeed, calculating distance matrices for many images (e.g., $90,000$) in the entire feature space ($l \times d = 524,288$) would have been unfeasible given our current computational resources. In section D of the Appendix, we show with an experiment on CIFAR10 that the ID of a representation decreases after average pooling. Still, the fundamental shape of the ID does not change qualitatively (see Fig. S3). In the same section, we also show that the ID profiles do not *quantitatively* change if we extract the representations after the attention maps or after the MLP blocks (see Fig. S4).

**Discussion of different ID shapes.** In Sec. 3.1, we showed that the ID profile in pLMs has three phases and terminates with a final expansion, while in iGPTs, there is a fourth phase where the ID is compressed in the last layers. These differences can be attributed to various factors. For instance, in the case of ImageNet, variations of the pixel brightness [13] and details of data preprocessing, including the substantial reduction of the input resolution (see Sec. A and [10]) can artificially alter the ID significantly. In the middle layers, instead, the ID is connected to the semantic complexity of the dataset, and its value tends to be consistent not only within transformers of the same class but also with the last hidden layer of convolutional networks in the case of the ImageNet dataset [13] and with other methods relying on different distance metrics in the case of proteins [30] (see Sec. 3.1). This implies that the ID of the compressed representations can be larger or smaller than the one measured in the early layers and close to the output, where the network reconstructs the inputs. These considerations may explain why the final layers of the network can either reduce the ID (as seen in iGPT) or expand it (as observed in ESM models). However, our key finding remains consistent in both cases: the layers where the ID reaches a local minimum correspond to representations where a semantic property of the data is most pronounced. Moreover, other important factors, such as the distinct training objectives (MLM in protein language models and next pixel prediction in iGPTs) and the different data modalities used for training, also influence the ID's characteristic shape.

**Preliminary exploration of NLP representations.** The NLP literature has previously discussed the idea that hidden representations in the middle layers of large transformer models encode abstract information that can be leveraged to solve various tasks successfully. Syntactic information is more prominent in the middle layers of deep transformers, as shown by syntactic tree depth reconstruction in [32]. However, the localization of semantic information has yielded contrasting results [33–37] possibly due to the complexity of language data and the ongoing evolution of model size and architectures. These studies show that language data requires diverse tasks and probes for comprehensive understanding. In addition, there is a substantial distinction between pLMs and Large Langue Models (LLMs). While for pLMs, we almost reached an overparameterization regime on the UniRef dataset [38], this saturation level has not been observed in the context of language. Recently, we started analyzing the geometry of hidden representation in the NLP domain with our methods. Our early analyses on GPT-2-XL on the SST dataset [39] did not reveal a second peak similar to the one in iGPT. With the release of the Llama models [40], we repeated the experiment on the Llama-v2 with 70 billion parameters (see Appendix, Sec. E). In this case, the evolution of the ID profile is more complex, showing three peaks and two local minima across the hidden layers. Remarkably, our key finding remains consistent: the highest overlap with class partition, determined by sentence sentiment, occurs in correspondence with the first local minimum. After the second peak, the ID profile shows a more complex behavior, which we are currently investigating. We will explore these aspects further in future work dedicated to NLP tasks.

## Acknowledgments and Disclosure of Funding

The authors acknowledge the AREA Science Park supercomputing platform ORFEO made available for conducting the research reported in this paper and the technical support of the Laboratory of Data Engineering staff. We thank Roshan Rao for kindly providing the weights of the ESB-2(650M) models at training checkpoints used to produce Fig. 3 (left).

A.A., A.C., and D. D. were supported by the project "Supporto alla diagnosi di malattie rare tramite l'intelligenza artificiale" CUP: F53C22001770002. A.A., A. C., and F.C. were supported by the European Union – NextGenerationEU within the project PNRR "PRP@CERIC" IR0000028 - Mission 4 Component 2 Investment 3.1 Action 3.1.1. L.V. was supported by the project PON "BIO Open Lab (BOL) - Rafforzamento del capitale umano" – CUP J72F20000940007.

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

# Appendix

## A Architecture details

Table 1: Characteristics of the models employed for extracting representations.

| Model | #Blocks | Emb. dim. | #Heads | #Params | Dataset | Reference |
|---|---|---|---|---|---|---|
| ESM-2 (35M) | 12 | 480 | 20 | 35M | UR50/D | [15] |
| ESM-2 (650M) | 33 | 1280 | 20 | 650M | UR50/D | [15] |
| ESM-2 (3B) | 36 | 2560 | 40 | 3B | UR50/D | [15] |
| iGPT-S(76M) | 24 | 512 | 8 | 76M | ImageNet | [10] |
| iGPT-M(455M) | 36 | 1024 | 8 | 455M | ImageNet | [10] |
| iGPT-L(1.4B) | 48 | 1536 | 16 | 1.4B | ImageNet | [10] |

**Transformer models for protein sequences.** The input data points, corresponding to proteins, are variable-length sequences of $l$ letters $s = a_1, a_2, \ldots a_l$, chosen from an alphabet of $n_a (\simeq 20)$ tokens corresponding to amino acids. Each token is encoded by an embedding layer into a vector of size $d$ so that the generic protein $s$ is represented as a matrix $x := f_0(x)$ of size $l \times d$. A model with $B$ blocks transforms a data point $x \in \mathbb{R}^{l \times d}$ into a sequence of representations: $f_0(x) \rightarrow f_1(x) \rightarrow \ldots \rightarrow f_B(x) \rightarrow f_{out}(x)$, where $f_i, i = 1, \ldots, B$ stands for the self-attention module at the $i^{th}$ block, and the final LM-head $f_{out}$ is a learned projection onto dimension $l \times n_a$. The size of each hidden layer does not change across the model and is equal to $l \times d$; therefore, the action of the model is a sequence of mappings $\mathbb{R}^{l \times d} \rightarrow \mathbb{R}^{l \times d}$. For each layer $i$, we choose to perform global average pooling across the row dimension $f_i(x) \rightarrow \frac{1}{l} \sum_{j=1}^{l} (f_i(x))_j$, since this reduction retrieves sufficient biological information to solve, directly or possibly after finetuning, homology, structural and evolutionary tasks. For a given pLM, the action of the network on a dataset of $N$ proteins can thus be described by $B + 1$ collections of $N$ vectors in $\mathbb{R}^d$: these are the data representations that we investigate.

**Transformer models for image processing.** The structure of the Image GPT (iGPT) transformers is very similar to that of pLMs. Due to the high memory footprint required by the attention layers, Chen et al. [10] reduce the image from the standard ImageNet size ($r = 224^2$) to $r = 32^2$ (S), $r = 48^2$ (M), $r = 64^2$ (L) and the three color channels are encoded in an "embedding axis" with size $d_{in} = 512$. In practice, the $\mathbb{R}^3$ color space in which each pixel is represented by a triplet of real numbers $(R, G, B)$ is quantized with $k$-means clustering ($k = 512$), and each pixel is described by the *discrete* "code" of the cluster where it belongs. An input image is then represented by a data point $x \in \mathbb{R}^{l \times d_{in}}$ where the $l$ pixels are placed along the sequence axis in raster order and $d_{in}$ encodes the color of each pixel. Like in pLMs, an input data point, after the embedding layers, is processed by a stack of attention blocks that produce a sequence of data representations of identical size $x \in \mathbb{R}^{l \times d}$. The final head $f_{out}$ projects the output of the last attention block to $\mathbb{R}^{l \times d_{in}}$, which encodes at each sequence position $i$ the conditional distribution

$$p(x_i) = p(x_i | x_0, \ldots x_{i-1}) \tag{1}$$

Once the network is trained, an image can be generated by sampling from $p(x_i)$ one pixel at a time.

# B  Details about the intrinsic dimension estimation.

Facco et al. [21] showed that the ID can be estimated using only the distances $r_{i_1}$, $r_{i_2}$ between each point $x_i$ and its first two nearest neighbors. Under the assumption of the Poisson point process in space [41], the ratios $\mu_i = r_{i_2}/r_{i_1}$ between the distances to the second and first nearest neighbor follow a Pareto distribution $p(\mu_i|d) = d\mu_i^{-d-1}$. Assuming that the dataset likelihood factorizes over the $N$ single-point likelihoods, the ID can be easily inferred given the empirical values of the $\mu_i$ as the $p(\mu_i|d)$ depend only on $d$. Facco et al. propose to estimate the ID from the cumulative distribution function $F(\mu) = 1 - \mu^{-d}$ with a linear regression fit. Indeed, the ID is the slope of the line through the origin obtained regressing $\log(\mu_{\sigma(i)})$ against $-\log(1 - F^{emp}(\mu_{\sigma(i)}))$ where $\mu_{\sigma(i)}$ are the $\mu_i$ sorted in ascending order and $F^{emp}(\mu_{\sigma(i)}) := \frac{i}{N}$ is the value of empirical cumulative distribution function at $\mu_{\sigma(i)}$ [21].

In practice, due to its very local nature, the TwoNN estimator can be affected by noise, which typically leads to an overestimation of the ID. To identify the most plausible value of the data intrinsic dimension, Facco et al. apply the TwoNN estimator on the full data set and then on random subsets of decreasing size: $n = [N, N/2, N/4, ...]$. The relevant data ID is chosen as the value of $\hat{d}$ where the graph $\hat{d}(n)$ exhibits a plateau, and the ID is less dependent on the scale. Recently, Denti et. al [42] proposed a better strategy to perform a multiscale analysis of the ID considering nearest neighbors of higher order instead of decimating the data set. This approach reduces the variance of the likelihood, giving more stable estimates.

These methods gave similar results in the experiments of this work: the ID estimated with the TwoNN was less scale-dependent across dataset sizes between $[N/2, N/8]$. Thus, in the main text, we report the TwoNN estimates on the dataset decimated by a factor of 4.

**Validity of the local density assumption.**   To quantitatively validate the constant density assumption, we employed the Point Adaptive kNN (PAk) method introduced by Rodriguez et al. [43]. PAk determines the extent of the neighborhood over which the probability density can be considered constant for each data point, subject to a specified confidence level. Applying PAk (as implemented in DADApy [25]) to our dataset revealed that, on average, the density can be considered constant within the first 6 neighboring data points. Our study measures the ID using the distances between a data point and its first two nearest neighbors. This analysis allows the conclusion that, at this scale, the assumption of local density holds.

## C   Intrinsic dimension and overlap profiles on other protein language models.

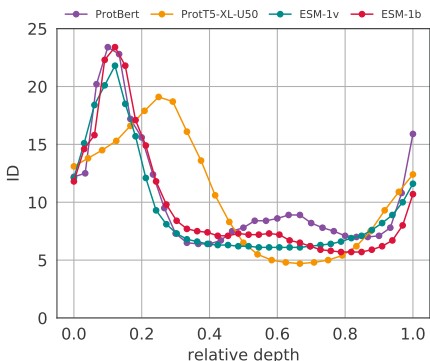

Figure S1: **The ID shape for different pLMs architectures.** The latest developments in the application of pre-trained pLMs for the solution of diverse biological tasks have been mainly fuelled by two families of models: Prot-Trans [7] and Evolutionary Scale Modelling (ESM) [6, 38, 44, 45]. During the last years pLMs with different architectures, number of parameters, and embedding sizes have been trained on several datasets obtained starting from the UniProt [46] database. The profiles in the figure complement the analysis done in Sec. 3.1, showing the ID curve of ProteinNet on pLMs trained on UniRef50 (ESM1b and ProtT5-XL-U50, red and orange) UniRef90 (ESM1v, green), Uniref100 (ProtBert, purple). Despite the significant differences of the pLMs considered in the analysis, the consistency of the three-phased behavior of the ID remains: an initial peak is followed by a plateau where the ID assumes low values, and the ID grows again to values close to the one measured after the positional embedding.

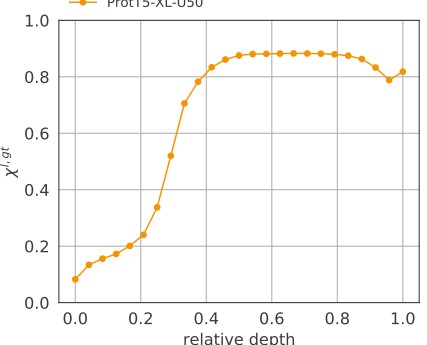

Figure S2: **Nearest neighbor search in plateau layers improves identification of protein relations.** It was recently shown in [47] that the first nearest neighbor searches for remote homologous protein domains based on the last hidden layer representations of the ProtT5-XL-U50 pLM outperform state-of-the-art methods based on sequence similarity. Adapting the approach of Sec. 3.2, we mimic the experiment performed in Sec. 2 of [47] by 1) considering protein domains in SCOPe belonging to a superfamily with at least 2 sequences, 2) setting the number of neighbors to $k = 1$. Considering representations in the plateau layer improves the accuracy of the 1-kNN homology search. In particular, in the figure, we observe an improvement of $\sim 6\%$ performing the search on a plateau layer instead of the last layer before the output. Importantly, the performance gain of $\sim 6\%$ is obtained without any further training.

# D Impact of the implementation details and hyperparameter choices.

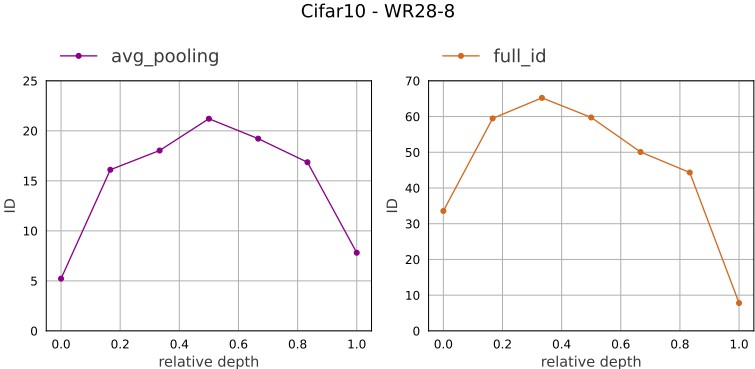

Figure S3: **Impact of the average pooling on the intrinsic dimension profiles.** To study how the average pooling affects the ID profile, we tested the representations of CIFAR10 in the Wide-ResNet28-8 model. The left panel shows the intrinsic dimensionality (ID) computed in the full feature space, while the right panel displays the ID after applying the global average pooling along the spatial dimensions (width and height). Notably, we observe a qualitative consistency in the shape of the ID profiles in both cases, as they conform to the typical bell-shaped curve characteristic of CNN architectures (see [13]). From a quantitative perspective, the ID profile after average pooling exhibits a downward shift as a consequence of the averaging procedure, particularly pronounced at the initial stages of the architecture. This effect is likely due to the low number of channels after the first block (16) compared to the later blocks (128, 256, and 512, respectively). Nevertheless, we can be confident about the qualitative robustness of the profiles as long as the number of "channels" (where channels are interpreted as embedding dimension) significantly surpasses the ID as in large transformer models. Indeed, in this case, the number of "channels" is substantially higher than in early CNN layers (ranging from 512 to several thousand in modern large language models) and remains constant throughout the architecture.

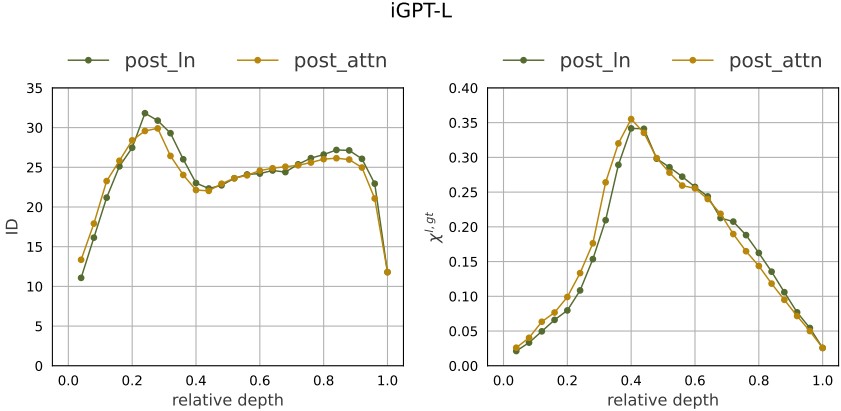

Figure S4: **Intrinsic dimension and neighborhood overlap after the self-attention blocks.** The figure compares the ID (left) and the overlap with the ground truth labels ($\chi^{l,gt}$, right) of the ImageNet representations after the first normalization layer with those of the representations after the attention maps of each self-attention block for the iGPT Large model. The ID and profiles are consistent, indicating the robustness of our analysis with respect to the layer choice.

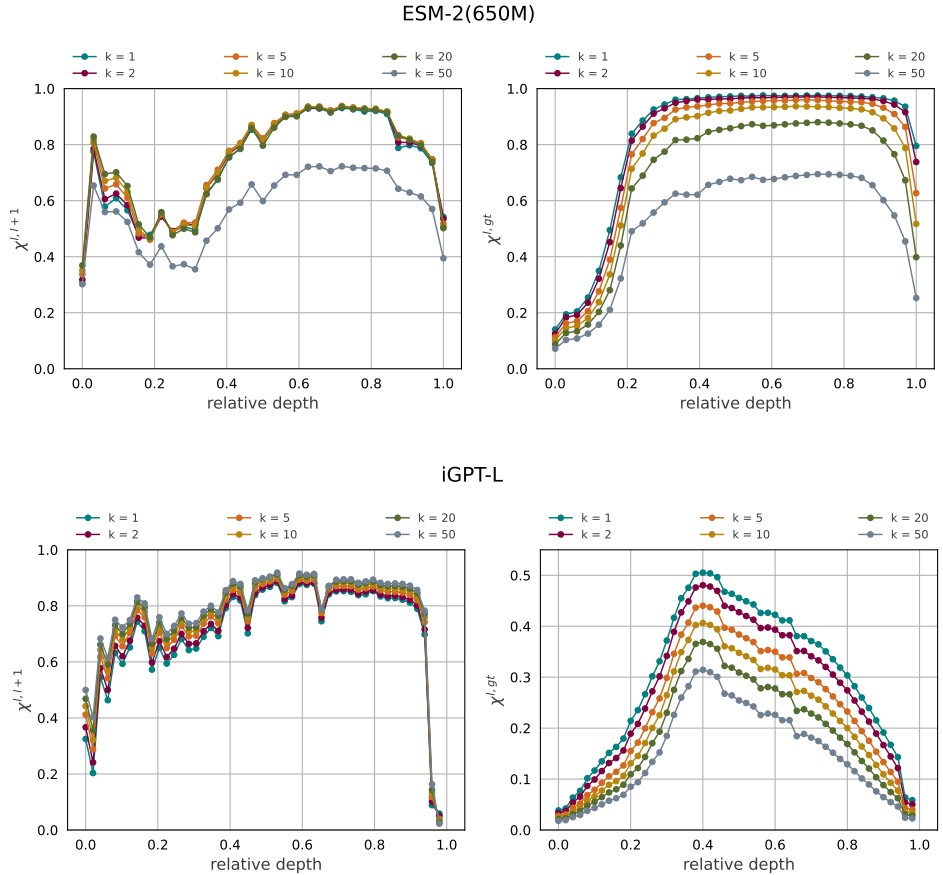

Figure S5: **Neighborhood overlap profiles are robust w.r.t. the choice of the neighborhood size $k$.** Overlap profiles in ESM-2 (650M, top) and iGPT Large (bottom) varying the neighborhood size $k$ in the range $\{1, 50\}$. The curves of the overlap between consecutive layers $\chi^{l,l+1}$ (left) and with the ground truth classes (right) are qualitatively unchanged. When considering the ESM-2 model (top), the lower values of $\chi_k^{l,l+1}$ and $\chi_k^{l,gt}$ when $k = 50$ is the consequence of certain superfamilies having fewer than 50 elements.

# E   Preliminary analysis of the intrinsic dimension and overlap in LLama2

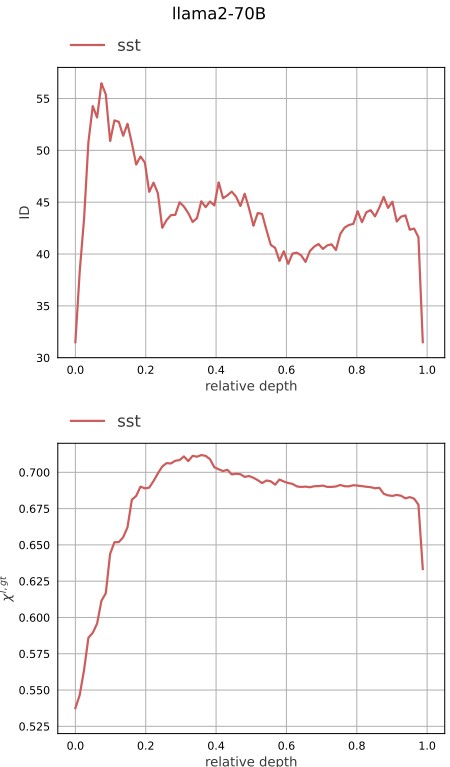

Figure S6: **Intrinsic dimension and neighborhood overlap of sst in LLama2-70b.** [Top] The intrinsic dimension (ID) of representations in hidden layers of Llama-v2 for the Stanford Sentiment Treebank (SST) dataset, plotted against block number normalized by the total number of blocks (relative depth). [Bottom] Overlap with ground truth labels, i.e. sentiment of the sentences, at each self-attention block, plotted against relative depth. In correspondence with the first local minimum of the ID profile, at 0.3 relative depth, the overlap with the class partition in the Bottom panel is the highest.

