# OpenReview forum: "The geometry of hidden representations of large transformer models"
_NeurIPS.cc/2023/Conference — NeurIPS 2023 poster_

### Official Review · Reviewer_wEGg · 2023-07-03

**Soundness:** 3 good
**Presentation:** 2 fair
**Contribution:** 3 good
**Rating:** 7
**Confidence:** 4

**Summary:**

This work investigates the geometry of hidden representations of transformer models trained via a self-supervised task on either amino acid prediction in proteins or pixel prediction in images. The work uses two tools to understand this geometry: intrinsic dimension (ID) (estimated via the TwoNN algorithm) and neighborhood overlap. The paper shows that as data passes through the layers of a transformer, both the ID and neighborhood overlap change in characteristic ways that the work connects with the extent to which the representation is organized by the semantic content in the data. The paper validates this claim by looking at the extent to which each data point shares neighbors that belong to the same semantic class, showing that this peaks at layers when ID is low and changes in neighborhood overlap is also low. The work then speculates that these observations could provide a way of identifying the best representations to use for downstream tasks.

**Strengths:**

- While there is a rich literature exploring hidden representations of deep learning models, most works continue to focus on CNNs or MLPs with supervised training on medium sized datasets. Given the growing importance of transformers to NLP and vision and the increasing use of self-supervision for large-scale training, there is a need for works that explore how these approaches impact a model’s hidden representations. Thus, this is a welcome work that will doubtless be of value to researchers training large transformers via self-supervision.
- The paper contains careful analysis of the experiments (as opposed to other works which all too frequently just list summary statistics). The conclusions which are reached are all fairly-well supported within the scope of the experiments that were performed. Crucially, the two metrics that are used, intrinsic dimension and neighborhood overlap, reinforce each other’s conclusions. This increases the believability of the results significantly.
- The paper is able to suggest some practical value (identifying layers of the transformer that capture the most semantic content) in their scientific observations (patterns in representation geometry in large-transformers), helping to connect practice with theory.


**Weaknesses:**

- The experiments in this work all use transformer models trained via self-supervised tasks based around data reconstruction (as opposed, for example, to a contrastive learning task). While most of the phenomena observed in this paper is attributed to the “large transformer” architecture, this reviewer wonders if some of the conclusions are also contingent on the self-supervision task. For example, does one see similar behavior in a large vision transformer that has been trained in a supervised manner. Disentangling which of the conclusions are due to architecture and which are due to training method would make the work significantly stronger.
- Many of the claims in the work rely on comparison between the shapes of curves which plot intrinsic dimension of data between layers. While the claims seem mostly reasonable given the figures, it would make the work stronger if more quantitative measures were used to, for example, compare ID curves. This sort of automation might also make it easier to compare against a broader range of models and datasets. While this reviewer thought that the use of both protein and image datasets was a strength of this work, comparing against other types of transformer architectures and datasets would reinforce the conclusions.
- The main content of the paper consists of discussion of several figures. This reviewer felt that the way this discussion was written/organized, it was easy for the reader to get disoriented and forget the main points already established. Possibly making the discussion more concise or better highlighting the main takeaways from each section would help the reader to mentally organize the primary findings of the work. Being more succinct might also allow further experiments to be included.

Nitpicks

- The abstract describes a transformer as being composed of a “sequence of functionally identical transformations”. To this reviewer, functionally identical transformations would be transformations that behave the same way on the level of functions (for the same inputs, they give the same outputs), but may be parametrized in different ways. The layers of a transformer are instead transformations that all belong to the same functional family, though each layer is generally a different function.
- Some sentences in this work have an over-abundance of commas. For example, “In this work, we systematically investigate, in some self-supervised models, fundamental geometric properties of representations, such as their intrinsic dimension (ID) and neighbor composition.” To improve the flow of the paper, it would be good to find such sentences and remove some of the commas.
- Line 66: “…where the annotation is very scarce” -> “…where the annotations are very scarce.”
- Reading this work would have been easier if the figures were closer to their corresponding analysis. As it is, the reader must flip between pages frequently to validate claims.
- Line 110: “More in detail,” -> “In more detail,”.


**Questions:**

- Given the success of transformers trained with self-supervised learning in the NLP domain, I was curious why the authors did not study this setting?
- Line 76: Is the assumption of locally constant density really “mild”? This reviewer wonders if image datasets, which have moderately high intrinsic dimension ([1] estimates ImageNet as having intrinsic dimension between 26 and 43) might be fairly sparsely sampled even when the dataset is large.
- What happens to the results when $k$ is varied (rather than being fixed at 30)?
- The paper says that hidden representations are extracted after the first normalization layer of the attention blocks. Do you have a sense of whether results change if you use representations from other parts of attention blocks?
- The results show the intrinsic dimension of the data increasing at times between layers. We know that mathematically, maps $f: M \rightarrow \mathbb{R}^n$ that map a $k$-dimensional manifold $M$ into $\mathbb{R}^n$ such that $\dim(f(M)) > k$ exist (e.g., space filling curves) but it is this reviewer's understanding that the space of such $f$ has measure zero with respect to many reasonable probability measures on the space of such maps. Given this, is it really possible that the intrinsic dimension of the representations is increasing? Or is their some other change that is changing the ID estimation?

[1] Pope, Phillip, et al. "The intrinsic dimension of images and its impact on learning." arXiv preprint arXiv:2104.08894 (2021).


**Limitations:**

This reviewer believes that some limitations could have been discussed, including:
- The use of intrinsic dimension estimators, which can provide misleading feedback in certain situations.
- A limited number of different model types and datasets.

---

> ### Author Rebuttal · Authors · 2023-08-09
>
> We thank reviewer wEGg for carefully reading our manuscript for providing several points of discussion that we address below.
>
> **Weaknesses**
>
> *[...] Disentangling which of the conclusions are due to architecture and which are due to training method would make the work significantly stronger.*
>
> We agree with the referee on this point. We will stress further in the final version of the manuscript that our analysis is specifically focused on transformer models trained by self-supervised reconstruction tasks (such as MLM or LM). We address the critical role of self-supervision in Appendix A.3.4: we show that self-supervised pre-training is crucial for the emergence of three-phased behavior by comparing our results with vision transformers trained for image classification. In that case, we can observe a much less pronounced second peak and a decrease of ID in the last layers to values lower than the first embedding.
>
> *[...] it would make the work stronger if more quantitative measures were used to, for example, compare ID curves.*
>
> It is true that we based our analysis of ID curves on the inspection of the figures: since the domain is one-dimensional this intuitive approach is particularly effective in this case. However, since the ID is measured on a small number of layers the local minimizers of the ID can be found easily by a brute-force search approach as well.
>
> *[...] comparing against other types of transformer architectures and datasets would reinforce the conclusions.*
>
> We kindly refer the reviewer to our response to Question 1.
>
> *This reviewer felt that the way this discussion was written/organized, it was easy for the reader to get disoriented and forget the main points already established.*
>
> We will take into consideration this advice for the drafting of the final version of the paper.
>
> **Questions**
>
> *I was curious why the authors did not study this (NLP) setting?*
>
> We kindly refer the reviewer to the global response about our preliminary investigation of NLP.
>
> *Line 76: Is the assumption of locally constant density really “mild”?*
>
> The reviewer is correct in highlighting potential issues in assuming locally constant density for datasets with high ID. To quantitatively validate this assumption, we employed the Point Adaptive kNN (PAk) method introduced by [2]. PAk determines, for each individual data point, the extent of the neighborhood over which the probability density can be considered constant, subject to a specified level of confidence. Applying PAk (as implemented in [3]) to our dataset  revealed that, on average, the density can be considered constant within the first 6 neighboring data points. In our study, the ID is measured using the distances between a data point and its first two nearest neighbors. This analysis allows concluding that, at this scale the assumption of local density holds.
>
> *What happens to the results when k is varied [...]?*
>
> We addressed the robustness of our findings concerning the hyperparameter k in Appendix A.3.4 (see Fig 6). Fig. 6 shows that the neighborhood overlap curves remain essentially unchanged for k<50  both between successive layers (Left) and with the ground truth labeling of the data (Right).
>
> *Do you have a sense of whether results change if you use representations from other parts of attention blocks?*
>
> In Fig.2 of the attached PDF we extended the analysis of the ID and the overlap with the ground truth labels ($\chi^{l, gt}$) on the representations after the first normalization layer and after the attention maps of each self-attention block for the iGPT model on ImageNet. The ID and $\chi^{l, gt}$ profiles are consistent with those shown in the manuscript which are extracted before each attention block, indicating the robustness of our analysis with respect to the layer choice. Due to time constraints in preparing this response, we perform the analysis only for iGPT-large. Nevertheless, we are confident that the observed trends would also hold true for smaller models.
>
> *[...] is it really possible that the intrinsic dimension of the representations is increasing? Or is there some other change that is changing the ID estimation?*
>
> The reviewer is indeed right in saying that from a mathematical standpoint the set of continuous functions $f: R^{d}\to R^{d}$ mapping a k-dimensional manifold $M\subset R^{d}$ onto a set that fills a manifold $M’$ with $dim(M’)>k$ is of measure zero.
> However, the manifold hypothesis, which is one of the working assumptions of our analyses, states that datasets (and their representations) typically lie “in proximity” of a smooth manifold often of much lower dimension than the embedding space. In particular, the ID approximates the number of independent coordinates that are necessary to describe the dataset without significant information loss [4]. When dealing with a finite set of data points the ID typically represents the number of dimensions where the dataset displays large/significant variations, neglecting those directions where the data variability is small/non-significant. Thus, the space of continuous (even smooth) functions that increase the intrinsic dimension in the sense of the manifold hypothesis is not of measure zero.
>
> **Limitations**
>
> *The use of intrinsic dimension estimators, which can provide misleading feedback in certain situations.*
>
> We will use the response to the reviewer’s concern about the local density assumption discussed above in the revised version of the manuscript.
>
> **References**
>
> [1] Touvron et al., “Llama 2: Open Foundation and Fine-Tuned Chat Models”, arXiv:2307.09288, 2023
>
> [2] Rodriguez et al., "Computing the free energy without collective variables, Journal of chemical theory and computation", 2018
>
> [3] Glielmo et al., "DADApy: Distance-based analysis of data-manifolds in Python, Patterns", 2022
>
> [4] Facco et al., "Estimating the intrinsic dimension of datasets by a minimal neighborhood information, Scientific Reports", 2017

---

> > ### Comment · Reviewer_wEGg · 2023-08-15
> >
> > I would like to thank the authors for their detailed response. It is especially interesting to see the initial results on language models and that the choice of $k$ does not strongly impact the findings. I have updated my rating to a 7. In general I think this is an interesting work and enjoyed the opportunity to read it.

---

### Official Review · Reviewer_kxbE · 2023-07-06

**Soundness:** 3 good
**Presentation:** 4 excellent
**Contribution:** 3 good
**Rating:** 6
**Confidence:** 4

**Summary:**

This paper presents an analysis of the internal representations of transformers trained with self-supervised learning, such as masked language modeling, from two perspectives: intrinsic dimension and adjacency structure. Experiments on two datasets - protein sequences and image data - revealed that the intrinsic dimension within the transformer has two peaks, one in the early layers and another in the latter layers. This result is qualitatively different from previous convolutional networks trained with supervised learning. Notably, the intermediate layer, where the intrinsic dimension is minimized, is the most suitable for categorical discrimination in downstream tasks.

**Strengths:**

This paper presents a notable analysis of the intrinsic dimension of transformer models trained with self-supervised learning. While many approaches have been used to analyze the internal dimension of deep neural networks, most have focused on convolution-based networks. The discovery of two peaks in the internal dimension is intriguing, offering insightful contributions to the community. The results are clearly presented, and the paper is well-structured, making it easy for readers to follow the logical development. Of particular interest is the finding that such representations can spontaneously appear in models trained with masked modeling, even without an explicit bottleneck structure in the model.

However, this interesting result also raises many considerations and discussions. There are points in the current paper where sufficient evidence to support the authors' claims is lacking and further room for discussion remains. These points will be discussed in detail below.

**Weaknesses:**

The paper's experimental comprehensiveness raises two issues. The first issue revolves around the focus on internal representations of transformer models trained with self-supervised learning, with the assertion that the evolution of the internal dimension displays two peaks. However, it appears that sufficient experiments have not been performed to identify the causes of this occurrence definitively. An ablation study would be incredibly beneficial in distinguishing the elements that stem from the model architecture (transformer) and those arising from the training protocol (masked/autoregressive modeling).

Secondly, the paper state in the introduction that the adjacency structure is “rearranged” at the peak of the representation’s internal dimension, regardless of the dataset domain. Yet, upon the qualitative comparison of Figures 1 and 2, it is challenging to confirm a consistent change in the internal dimension and the overlap of the adjacency structure between layers in the image domain experiments. Therefore, the current portrayal of the contributions gives the impression of over-claiming. As asserting a clear correlation in qualitative behavior is difficult, a comparison with some control conditions should be made. Additionally, it would be desirable to have a discussion concerning the influence of differences in the data domain.

**Questions:**

I have several questions for the authors:

1. In interpreting Figure 1, the main text mentions that the changes in the internal dimension qualitatively match the experiments for the two domains. However, there seems to be a qualitative difference between domains concerning whether there is concentration or expansion near the final layers in the latter half. What could be the reasons for such a difference?
2. Regarding the analysis of the overlap of the adjacency structure, the paper mentioned that the authors used $k=10$ and $k=1$ in the protein domain experiments, and $k=30$ in the image domain experiments. Generally, the experimental results would depend on this hyperparameter. How robust are the authors' claims with respect to these hyperparameters, and what is the rationale or justification for the chosen hyperparameter values?
3. This paper mentioned that the authors used the GAP type averaging procedure along the token direction when analyzing the internal representation of the transformer model. While this is not a problem when comparing results within the same model architecture, it seems caution is needed when comparing the results with other models. For instance, when comparing the results with convolution-type neural networks, how can we guarantee the qualitative robustness of the statements depending on whether this averaging operation is performed or not?

**Limitations:**

Given that this paper is not proposing a new method but focusing on analyzing the internal representation of existing methods, it may not strictly apply to a discussion on limitations. The estimation of the internal representation is based on a method proposed in previous research, and the applicability of this method has been sufficiently demonstrated. However, as mentioned in the Weakness and Question sections, there are concerns about the comprehensiveness of the experiments in this paper. Improvements in these points would lend greater significance to the authors' claims.

---

> ### Author Rebuttal · Authors · 2023-08-09
>
> We thank reviewer kxbE for carefully reviewing our manuscript and for the constructive comments.
>
> **Weaknesses**
>
> *An ablation study would be incredibly beneficial in distinguishing the elements that stem from the model architecture (transformer) and those arising from the training protocol (masked/autoregressive modeling).*
>
> A specific ablation study is described in Appendix A.3.4, Fig. 7, where we examined the vision transformers (ViT) trained for an image classification task. In this case, the ID profile exhibits a much less prominent second peak followed by a gradual decrease of ID below initial values, consistent with the established pattern typically observed in image classifiers [1]. This example shows that the shape of the ID profiles is influenced by the training protocol and is not an inherent characteristic of the transformer architecture itself. We are investigating how-fine tuning affects the models we analyzed in the manuscript. However, it is crucial to emphasize that the primary focus of this work is on studying ID profiles in models trained through self-supervision. More importantly,  we aimed to establish connections between the low-dimensional representations and the layers where the semantic properties of the data are better expressed.
>
> *[...] upon the qualitative comparison of Figures 1 and 2, it is challenging to confirm a consistent change in the internal dimension and the overlap of the adjacency structure between layers [...] the current portrayal of the contributions gives the impression of over-claiming. [...] Additionally, it would be desirable to have a discussion concerning the influence of differences in the data domain.*
>
> We would like to remark that the value of chi reported in Fig. 2 of the manuscript refers to the overlap between consecutive layers. The average value of chi in the first part of the network is of order 0.5 in the case of protein sequences and of order 0.7 in the case of images. This implies that after ~10 layers the overlap with the input is of 0.1% in the case of protein sequences and of  ~3% in the case of images. We believe that this can be called a significant rearrangement of the neighborhood, at least in the case of proteins. In order to avoid over-claiming we removed the word “profoundly” before “rearranged” in the introduction and in the text, since in the case of images an overlap of ~ 3% after the peak might indicate the survival of some faint memory of the input. We will also state explicitly that the rearrangement is more significant in the case of protein sequences highlighting the differences in the two data domains in this respect.
>
> **Questions**
>
> *However, there seems to be a qualitative difference between domains concerning whether there is concentration or expansion near the final layers in the latter half. What could be the reasons for such a difference?*
>
> We kindly refer the reviewer to the global response about the qualitative difference of ID profile between iGPT and ESM models.
>
> *How robust are the authors' claims with respect to these hyperparameters [k=1, k=30], and what is the rationale or justification for the chosen hyperparameter values?*
>
> We chose k=1 for pLMs to show how nearest neighbor search in plateau layers improves identifying protein relations. Other k values (k=10, k=30) do not affect the observations presented in our manuscript. We addressed the robustness of our findings concerning the hyperparameter k in Appendix A.3.4. In particular, Fig. 6  shows that the neighborhood overlap curves remain essentially unchanged for k<50  both between successive layers (Left) and with the ground truth labeling of the data (Right).
>
> *This paper mentioned that the authors used the GAP type averaging procedure [...] when comparing the results with convolution-type neural networks, how can we guarantee the qualitative robustness of the statements depending on whether this averaging operation is performed or not?*
>
> We emphasize that the global average pooling (GAP) approach was not intended for comparing our results regarding the geometric properties of transformer representations with the characteristics of hidden layers in convolutional architectures.
> However, to directly address the concern raised by the reviewer regarding the robustness of the GAP procedure in convolutional neural networks (CNNs), we conducted a test on the representations of CIFAR10 in the Wide-ResNet28-8 model [2]. We describe the results in Fig. 4 of the attached PDF. The left panel shows the intrinsic dimensionality (ID) computed in the full feature space, while the right panel displays the ID after applying GAP along the spatial dimensions (width and height). Notably, we observe a qualitative consistency in the shape of the ID profiles in both cases, as they conform to the typical bell-shaped curve characteristic of CNN architectures [1]. From a quantitative perspective, the ID profile after GAP exhibits a downward shift as a consequence of the averaging procedure, particularly pronounced at the initial stages of the architecture. This effect is likely due to the low number of channels after the first block (16) compared to the later blocks (128, 256, 512 respectively). Nevertheless, we can be cautiously confident about the qualitative robustness of the profiles as long as the number of “channels” (where channels are interpreted as embedding dimension) significantly surpasses the ID as in large transformer models. Indeed, in this case the number of "channels" is substantially higher than in early CNN layers (ranging from 512 to several thousands in modern large language models), and remains constant throughout the architecture.
>
> [1] Ansuini et al., “Intrinsic dimension of data representations in deep neural networks”, 32nd Conference on Neural Information Processing Systems, 2018
>
> [2] S. Zagoruyko and N. Komodakis, “Wide Residual Networks”, Proceedings of the British Machine Vision Conference 2016

---

> > ### Comment · Reviewer_kxbE · 2023-08-22
> > **Response**
> >
> > Thank you for your feedback on my questions and concerns.
> > The feedback from the authors' addressed all of my questions: qualitative difference in the domain, robustness on hyperparameters, use of GAPs. Based on their responses, I would like to increase the score to 6.

---

### Official Review · Reviewer_4o8z · 2023-07-06

**Soundness:** 4 excellent
**Presentation:** 4 excellent
**Contribution:** 4 excellent
**Rating:** 9
**Confidence:** 4

**Summary:**

This work focuses on characterizing the geometrical and statistical properties of data representations across the hidden layers of large transformer models. Specifically, it demonstrates the similarity in the evolution of geometric properties, such as the intrinsic dimension, between image-based and protein-based transformers. Additionally, the work proposed an intuitive, unsupervised strategy to identify the most semantically rich representations of transformers. The effectiveness of this strategy is demonstrated by the SOTA performance achieved by leveraging previous work and replacing the last layer with the layer identified in this study.

**Strengths:**

- This work presents a novel strategy to select the layers that produce the most semantic representation to be used in downstream tasks, opening many possible applications. Moreover, the method seems to be general and could be exploited in other architectures where the layer choice is currently arbitrary, such as classifiers.

- This work demonstrates state-of-the-art results in identifying protein relations by leveraging existing methods. It simply involves swapping the previously employed last layer with the layers that maximizes the semantic content identified through the strategy proposed in this work (Figure 5, right).

- The writing is extremely clear and easy to follow, presenting intuitive ideas, a solid experimental setup and convincing results.

- This work does not involve training models but instead analyzes publicly available models, completely avoiding any possible bias that could have been introduced in the training process.

- Among the others, the insight that large transformers behave essentially like sophisticated autoencoders (lines 337-338) is insightful and may serve as a source of ideas for future research.

**Weaknesses:**

This work presents a novel idea with exceptional clarity in its writing, accompanied by a robust experimental setup that yields convincing results. While no major weaknesses were identified, a more comprehensive discussion on the current limitations of the study would enhance its overall quality even further.

Minor:

- In line 36 (and others), the paper mentions the term "semantically rich representation." However, it is important to clarify how this term is defined. It can be argued that the richness of semantic content in a representation is dependent on the specific downstream task being addressed. In other words, a representation can be considered more or less semantically rich based on its effectiveness in a particular downstream task.

- To enhance readability, it would be beneficial to include the ID (or at least the minimum ID) directly in the plot in Figure 4. This additional information (even though repeated from Figure 3), would facilitate the reader to compare the peak categorical information with the ID minimum.

**Questions:**

- In lines 131-132, the paper mentions a method to compress the resolution and color channels of images to address memory constraints. However, it raises the question of why a methodology similar to the one described in [1] was not adopted. In [1], a "first-stage" autoencoder was trained specifically to compress the images into lower-dimensional latent representations while maintaining perceptual equivalence (and kept frozen in subsequent stages). Is there a specific reason why this approach was not considered in the current work?

- In line 286, it is reported that the plateau extends beyond half of the layers. This raises the question of whether this observation could be an indication that the model is over-parameterized or, at the very least, has more layers than necessary.  Further investigation or discussion could shed light on this question and provide valuable insights in the minimum network complexity to produce representations with the same semantic information.


[1] Rombach, Robin, Andreas Blattmann, Dominik Lorenz, Patrick Esser, and Björn Ommer. “High-Resolution Image Synthesis with Latent Diffusion Models.”, Proceedings of the IEEE/CVF Conference on Computer Vision and Pattern Recognition (CVPR), 2022


**Limitations:**

Although not extensively, Section 4 explores the limitations and potential areas for future research.

---

> ### Author Rebuttal · Authors · 2023-08-09
>
> We express our gratitude to reviewer 4o8z for their appreciation of our work, for carefully reviewing our manuscript, and for the stimulating comments.
>
> **Weaknesses**
>
> *In line 36 (and others), the paper mentions the term "semantically rich representation." However, it is important to clarify how this term is defined. It can be argued that the richness of semantic content in a representation is dependent on the specific downstream task being addressed. In other words, a representation can be considered more or less semantically rich based on its effectiveness in a particular downstream task.*
>
> We agree with this observation, and we will clarify that the expression "semantically rich representation" is relative to a specific task (typically of classification).
>
> *To enhance readability, it would be beneficial to include the ID (or at least the minimum ID) directly in the plot in Figure 4.*
>
> We thank the reviewer for this suggestion. We will consider adding an explicit indication of the minimum ID in the plot of Fig. 4 in the revised version of the manuscript.
>
> **Questions**
>
> *In lines 131-132, the paper mentions a method to compress the resolution and color channels of images to address memory constraints. However, it raises the question of why a methodology similar to the one described in [1] was not adopted.*
>
> The resolution and color channel compression mentioned in Line 131 of the Manuscript are part of the original encoding procedure of the iGPT models developed by [2], and thus they are not our decision. Our work is just focused on the analyses of transformer models developed by other groups and trained by self-supervision. In the discussion section, we will highlight that an examination of the approach by [1]  is left for future research.
>
> *In line 286, it is reported that the plateau extends beyond half of the layers. This raises the question of whether this observation could be an indication that the model is over-parameterized or, at the very least, has more layers than necessary.*
>
> This is a very interesting and pertinent point. Indeed, we recently started exploring the possibility of eliminating or replacing by a suitable transformation certain layers in this part of the network without compromising the performance. We will highlight this as a potential avenue for future research in the discussion section.
>
> **References**
>
> [1] Rombach et al., “High-Resolution Image Synthesis with Latent Diffusion Models”, Proceedings of the IEEE/CVF Conference on Computer Vision and Pattern Recognition (CVPR), 2022
>
> [2] Chen et al., “Generative Pretraining From Pixels”, 37th International Conference on Machine Learning, 2020

---

> > ### Comment · Reviewer_4o8z · 2023-08-13
> >
> > Thanks for the rebuttal! It addresses the comments and questions raised. I do not have any further questions, and I confirm my rating.

---

### Official Review · Reviewer_K92u · 2023-07-08

**Soundness:** 3 good
**Presentation:** 3 good
**Contribution:** 3 good
**Rating:** 5
**Confidence:** 1

**Summary:**

The paper studies the hidden representation of pretrained transformers via the ID (intrinsic dimension) on protein language tasks and image reconstruction tasks . The papers show that on protein LM, from input to output layer, the ID first increase to a peak, then decrease to an elbow, and finally increase to near its ID value at input layer; this is robust from small to largest models. For image reconstruct, the picture is quite different; the input & output layers have the smallest ID and there two peaks near the input and output layers, respectively. The results on iGPT is less robust compare to protein LM; (e.g., small iGPT does not seem to have two peaks).


Overall, I think the paper is well written, presentations and experimental setup are clean, the results are interesting. However, I am not very familiar with the field and could not judge the novelty of the paper.





**Strengths:**

A well written paper; structure is clean; presentation is good; authors spent decent amount of efforts to introduce / motivate / explain key concepts using in the paper, e.g. ID etc.

The results seem interesting even to folks who are not working this area.








**Weaknesses:**

The results on iGPT is less robust (compare to protein LM). In particular, for the small model (I don’t see two peaks). Naively, I also expect the shape of ID of iGPT to be similar to pLM, encode (decreasing ID, smaller than input) and decode (increasing ID), just like the elbow in the pLM. Is there an explanation?

Why not also including a third task: NLP  (casual language model) in the main text?


**Questions:**

See above.

**Limitations:**

Have several discussion about further extension of the current approach.

---

> ### Author Rebuttal · Authors · 2023-08-09
>
> We thank reviewer K92u for reviewing our manuscript.
>
> **Weaknesses/Questions**
>
> *The results on iGPT is less robust (compare to protein LM). In particular, for the small model (I don’t see two peaks).*
>
> We agree that in the small model the second peak is absent. However, the small model is significantly less accurate than the larger model, both in terms of validation loss (refer to Fig. 3, [1]) and performance in image classification tasks, as evaluated through linear probes (see Fig. 3, [1]) and neighborhood overlap (Fig. 4, right panel, in our work).
> We emphasize that also in the case of protein language models (pLMs), the smallest model  (ESM2-8M, see attached PDF Fig. 1)  does not exhibit a clear three-phased ID profile (no peak and plateau), and the three-phased behavior becomes gradually more pronounced when increasing the size of the models, similarly to the case of iGPT models. Indeed large transformer models can develop qualitative changes in their performance and internal workings as their size is scaled up, so that a feature that is absent in smaller models can suddenly appear as the number of parameters is increased [2].
>
> *Naively, I also expect the shape of ID of iGPT to be similar to pLM, encode (decreasing ID, smaller than input) and decode (increasing ID), just like the elbow in the pLM. Is there an explanation?*
>
> We kindly refer the reviewer to the global response about the difference of ID profile between iGPT and ESM models.
>
> *Why not also including a third task: NLP (casual language model) in the main text?*
>
> We kindly refer the reviewer to the global response about our preliminary investigation of NLP.
>
> **References**
>
> [1] Chen et al., “Generative Pretraining From Pixels”, 37th International Conference on Machine Learning, 2020
>
> [2] Wei et al. “Emergent Abilities of Large Language Models”, Transactions on Machine Learning Research, 2022

---

> > ### Comment · Reviewer_K92u · 2023-08-15
> > **Re**
> >
> > Thanks for the detailed response, in particular, for running new experiments on the 70B llama-2 model. I will keep my score.

---

### Author Rebuttal · Authors · 2023-08-09

We thank all reviewers for the detailed and diligent reading of our paper, from which we took a lot of cues on how to improve the quality of our work. We would also like to express our gratitude for the general appreciation of our contribution. We reply to common points raised by some Reviewers here below.

**We remind everyone that the PDF file containing the figures supporting our responses is attached to this post.**

*Reply to the concern raised by reviewers **K92u** and **kxbE** regarding the difference in the final part of the ID profiles between iGPT and ESM.*

We agree with the reviewers that there is a qualitative difference in the last part of the ID profiles. In the main text, we mention that the “ three phases can be recognized in all the transformers and a fourth phase which we observed only in the iGPT models” (lines 172-173). We also emphasize that  “In the last part of the network [iGPT], at variance with what is observed in pLMs, the ID does not remain constant but grows again (more moderately) forming a second shallow peak almost at the end of the network” (lines 219-220-221).
Indeed, the ID in the middle part of the architecture need not be smaller than that at the input. As we state in the manuscript, the ID in the middle layers of the architecture is connected to the semantic complexity of the dataset, which is different in the two datasets. More precisely, in the context of protein sequences, Facco et al. (2019) show that the ID of a family of proteins, which measures the amount of variability arising from evolutionary changes in protein sequences, is between 6 and 12. Conversely, when dealing with images, the ID related to the semantic complexity of the dataset could be roughly measured from the values at the output of the state-of-the-art classifiers, where the representation is most compact. For instance, in the case of the ImageNet dataset, the ID at the output of various ResNet CNN is within the range of 15-20 (Ansuini et al., 2019). These values are remarkably consistent with those observed in our study: 5-6 for ESM-2, and 22 for iGPT on ImageNet (see Fig. 1 in the manuscript).\
On the contrary, the ID at the first layer of the network is influenced by various aspects of the raw input data such as the brightness of the input pixels (Ansuini et al., 2019) or other factors, and can be either larger or smaller than the ID measured at middle layers of the network. Since in the output layer the network tries to reconstruct (part of) the input data, these considerations can explain why there can be concentration (iGPT) or expansion (ESM) in the final layers of the network.
However, the *key finding of our study remains consistent in both cases*: the layers where the ID reaches a local minimum correspond to representations in which the overlap with a semantic property of the data is most pronounced.

*Reply to the question of reviewers **K92u** and **wEGg** regarding the potential extension of the current work to the NLP domain.*

At the time of submission, the transformers we analyzed, trained on Natural Language Processing (NLP) tasks, were not large enough to observe a second peak akin to the one in the manuscript for iGPT. In the Appendix, we reported the estimate of the intrinsic dimension (ID) of a GPT-2-XL model on the Stanford Sentiment Treebank (SST) dataset (see Fig. 8 in the Appendix), and commented on our findings.\
Recently, we conducted the same experiment on the much larger Llama-v2 model (70B parameters) that was released last July (Touvron et al., 2023). In this latter case, described in Fig. 3 of the attached PDF, our preliminary analysis shows that the evolution of the ID profile is more complex: after the first peak, the ID increases again in the middle of the architecture.
Remarkably, despite the dataset being quite different from ImageNet and Uniref, the *key finding of our present work remains consistent in the NLP case as well*: in correspondence with the first local minimum at one-third of the architecture depth (0.3 relative depth) the overlap with the class partition (given by sentiment of the sentence)  is the highest. \
After the second peak, the ID profile shows a more complex behavior. We believe that a thorough analysis of this and other downstream tasks is necessary to fully decipher the meaning of such profiles. We are excited to clarify and extend these aspects in a follow-up of this work, dedicated to NLP tasks.

**References**

Facco et al., “The intrinsic dimension of protein sequence evolution”, PLOS Comp. Bio., 2019

Ansuini et al., “Intrinsic dimension of data representations in deep neural networks”, 32nd Conference on Neural Information Processing Systems, 2018

Touvron et al., “Llama 2: Open Foundation and Fine-Tuned Chat Models”, arXiv:2307.09288, 2023

---

### Decision · Program_Chairs · 2023-09-21

**Decision:**

Accept (poster)

**Comment:**

While there were initially concerns on the scope of the experiments and generality of the results, the rebuttal results and discussion addressed these points, and the paper should be of general interest to the NeurIPS community. I encourage the authors to take all of the reviewer feedback into account when preparing the final version.